# Determining Trends and Factors Associated with Self-Reported Physical Activity among Adolescents in Rural North Carolina

**DOI:** 10.3390/ijerph191811417

**Published:** 2022-09-10

**Authors:** Sina Kazemzadeh, Chloe E. Opper, Xiangming Fang, Suzanne Lazorick

**Affiliations:** 1Brody School of Medicine, East Carolina University, Greenville, NC 27834, USA; 2ECU Health Residency Program, Pediatrics, Greenville, NC 27834, USA; 3Department of Biostatistics, College of Allied Health, East Carolina University, Greenville, NC 27834, USA; 4Departments of Pediatrics and Public Health, Brody School of Medicine, East Carolina University, Greenville, NC 27834, USA

**Keywords:** health disparities, community-based research, environment, obesity, physical education

## Abstract

It is important to better understand factors associated with physical activity (PA) levels in adolescents in rural areas. Cross-sectional data were used to obtain self-reported PA levels among adolescents in a school-based intervention in fall 2018. Demographic data, environmental variables, and cardiovascular fitness (PACER score) were also measured. Analyses included a two-sample *t*-test, ANOVA, a Chi-square test_,_ and a linear regression model. Participants included 3799 7th graders. Male (*p* < 0.0001), white (*p* < 0.0001), and healthy weight (*p* < 0.0001) participants reported more days of PA. The correlation between school physical education (PE) and PACER was modest (r = 0.27, *p* < 0.0001). Multiple linear regression model showed significant effects of school PE (*p* = 0.0011), gender (*p* < 0.0001), race (*p* < 0.0001), and weight category (*p* < 0.0001) on self-reported PA. The percentage of students reporting 60 min of PA for 5 (*p* < 0.0001) or 7 (*p* = 0.0307) days per week tended to be higher with increased days per week of school PE. Policy changes that increase PA and PE in middle schools may present opportunities to improve PA levels in adolescents, with emphasis on being inclusive and mindful of minority and female youth.

## 1. Introduction

Adolescence serves as a critical period during development for children to establish healthy lifestyle choices. Adolescents who participate in more physical activity (PA) tend to have stronger bones, stronger muscles, higher levels of cardiorespiratory fitness, and lower body fat [1]. Children and adolescents who are regularly active have a lower likelihood of developing chronic diseases, such as heart disease, hypertension, and type 2 diabetes as adults [1].

The current recommended guideline for children and adolescents between ages 6 to 17 years is at least 60 min of daily moderate-to-vigorous physical activity (MVPA) [1]. In the United States, less than a quarter (24%) of children ages 6 to 17 get 60 min of MVPA every day [2]. Additionally, adolescents in minority groups and who live in rural settings are more likely to be less physically active and overweight or obese [3,4]. Socioeconomic status, limited number of recreational facilities, parks, walkable roadways, and overall insufficient environmental support towards PA are likely contributors [3,4].

School-level factors, such as recess and physical education (PE), may also play an important role in adolescent PA levels and overall health. A systematic review by Gray and colleagues found that longer duration of recess and meeting recommended recess and PE time were significantly associated with lower rates of obesity [5]. However, they suggested the importance of further study on this topic due to the small number of studies and variability of measures used in their analysis [5]. A study in Canada by Ploeg and colleagues focused on whether a school-based intervention called comprehensive school health (CSH) can increase adolescents’ PA levels at school and outside of school [6]. The program targeted socioeconomically disadvantaged areas and educated students on how to improve upon healthy living habits [6]. At the conclusion of the two year study, the increase in mean steps was significantly greater in CSH intervention schools than in comparison schools [6]. Another study by Nader and colleagues of elementary school students in rural Oregon found significant associations between students’ PA levels and the amount of school PE and positive teacher attitude towards PA [7]. They also concluded that further investigations are needed between school-level PA factors, such as PE, and students’ PA levels at rural schools [7].

An Australian study examined the effect of a school-based intervention called Fit-4-Fun on PA levels and other health outcomes [8]. Fit-4-Fun was an eight week program that included health and PE lessons, a break time activity program (recess and lunch), and a home fitness program [9]. The study concluded that the students who participated in the program recorded more PA on the pedometer than the control group at three and six months post-intervention [8]. This shows sustained change even after the program was completed, highlighting the importance of intervention in the school setting. Another study in China by Li and colleagues found that children and adolescents who participated in a similar 12 week, school-based PA intervention had a statistically significant increase in in-school moderate to vigorous physical activity (MVPA) compared to the control group [10]. Taken together, the findings of these studies suggest that increased PA opportunities and PE at schools can lead to increased PA levels among students. However, it is still not known for adolescents what school and community factors can positively influence PA levels, especially in rural areas. Barriers to PA are present at a school and community level, particularly in these areas with less environmental support towards PA. If positive factors can be identified, steps can be taken to address the disparities that are hindering adolescents from achieving recommended PA levels.

To address this gap, we used existing data from a school-based wellness intervention called Motivating Adolescents with Technology to Choose Health^TM^ (MATCH) from the 2018–2019 school year to evaluate demographic, environmental, and school-level factors that may impact self-reported PA in 7th grade students attending middle schools across rural, eastern North Carolina. The majority of the participating schools are located in the rural, eastern part of the state, which is associated with a high risk of obesity and poor health in the youth population, which may be partly due to low environmental support towards PA at schools in these areas [4]. The theoretical model used to develop this research study was the Ecological Model of Active Living [11]. In this study, we evaluated the relationship between school- and community-level factors and self-reported PA levels in adolescents across rural North Carolina. We hypothesize that adolescents who attend schools with more PE and PA opportunities, and in areas with more exercise opportunities, will report participating in more PA.

## 2. Methods

In this cross-sectional study, we investigated the relationship between school PE, school PA, and exercise opportunities in the communities of the participating schools with students’ self-reported PA levels at the start of the school year. Other participant-level variables evaluated included sex, race, weight category determined from body mass index (BMI) percentile, and the progressive aerobic cardiovascular endurance run (PACER) score (a measure of cardiovascular fitness).

### 2.1. Study Setting and Participants

The MATCH participants for this study consisted of 7th graders from 40 middle schools across 14 rural North Carolina counties. The MATCH initiative utilizes social cognitive theory (SCT) to integrate lessons on healthy lifestyle habits into the regular 7th grade curriculum, and details on the MATCH intervention have been described previously [12,13,14]. The MATCH intervention has expanded each year since its inception in 2006, and is now in place in 70 schools as of 2022. Demographic characteristics of participants (sex, race/ethnicity) were collected at the beginning of the 2018 school year using student information provided at the time of school registration [13]. All 7th grade students who were enrolled in mainstream classes in the MATCH schools received the curriculum and were eligible for this study.

### 2.2. Measures

#### 2.2.1. Dependent Variable—Self-Reported PA

Lifestyle habits were assessed through the self-reported Sleep, Eating, Activity, and Technology (SEAT) questionnaire developed for use in MATCH [13]. As a part of the SEAT questionnaire, participants answered questions on the amount of PA they perform on a weekly basis using the following question from the validated Youth Risk Behavior Surveillance System (YRBSS) [15]. The question asks, “How many days each week are you physically active for at least 60 min each day? Add up all the time you spend doing any kind of physical activity that increases your heart rate and makes you breathe harder.” Students selected one answer choice for each question that ranged from 0 to 7 days a week. The results were used in the following two ways for analyses: as a continuous variable to determine the average number of days per week that the students achieved 60 min of PA, and also dichotomized as reported in YRBSS results into the percentage of students who met the thresholds of at least one, five, or seven days per week; of note, these categories are not mutually exclusive. The latter variable is reported by the Centers for Disease Control and Prevention (CDC) YRBSS, as part of monitoring health-related behaviors that are among the leading causes of death and disability among the youth, with “less risk” being assigned to reporting more days per week of PA [15]. Whether or not participants reported taking part in at least 60 min of PA at least one, five, or seven days of the week based on the results of the questionnaire was used as the dependent variable in this study.

#### 2.2.2. Independent Variables

As described in a previous publication by Opper et al, county- and school-level determinants were measured to assess PA opportunities for each study participant based on the school attended [16]. Variables were selected based on a review of the literature and expert consultations focusing on the social determinants of health and obesity in adolescents [16]. Each variable was assessed based on pre-determined standard criteria and then scored on a 5-point scale (Table 1) [16]. For the school-level determinants, school administrators completed questionnaires and a 5-point rating was applied to possible responses. Each school was then assigned a score between 1 to 5 for each variable, with a score of 1 being an unfavorable score and a score of 5 being the most favorable score [16].

#### 2.2.3. Access to Exercise Opportunities

Access to exercise opportunities served as a county-level determinant in MATCH participants. The data for this variable were obtained from the County Health Rankings and Roadmaps [17]. This variable measured the “percentage of population with adequate access to locations for physical activity”. Additional descriptions, a rating scale, and a source for this determinant are shown in Table 1.

#### 2.2.4. Physical Education and Activity

School administrators responded to two questions that asked about physical education (PE) and PA opportunities at their schools. These questions asked, “How is health/PE provided for 7th grade in mainstream classes?” and “Is there any other time during the school day, other than PE, that students are able to and regularly participate in physical activity (ex. recess)?” The questions, answer choices, and ratings can be seen in Table 1.

#### 2.2.5. PACER

As part of the MATCH program, students completed the Progressive Aerobic Cardiovascular Endurance Run (PACER) test at baseline and post-intervention as a measure of fitness. The PACER test measures the aerobic capacity and health-related components of fitness of more than 10 million students across the United States in over 20,000 schools [18]. Students are scored by the number of completed laps within the designated test time frame (for example, 25 completed laps equals a score of 25). Each lap is 20 m and students must complete each lap by the time a beep sounds [19]. The pace gets faster each minute, making the test more difficult as time goes on. During the first minute, participants are allowed nine seconds to run the distance, the lap time decreases by approximately one-half second after each minute [19].

#### 2.2.6. Body Mass Index

Participants’ anthropometric measures (height and weight) and lifestyle habits were assessed at the beginning and end of the 2018–2019 school year; the baseline measures were used for this cross-sectional study [13]. All students had their height and weight measured privately by trained personnel using a stadiometer and calibrated scale [13]. Then, BMI was calculated from height and weight. Sex-specific BMI z-score (zBMI), BMI percentile, and weight category (underweight <5th percentile; healthy weight 5th percentile–<85th percentile; overweight 85th–<95th percentile; obese >=95th percentile) were determined using standardized Centers for Disease Control (CDC) parameters [20].

### 2.3. Data Analysis

Data were summarized with descriptive statistics (frequencies for categorical variables and mean and standard deviation for quantitative variables). A two-sample *t*-test and analysis of variance (ANOVA) were used to access the bivariate associations between self-reported days of 60 min of PA per week and sex, race, and weight category. Pearson correlations (or Spearman correlations as appropriate) were used to access the bivariate associations between self-reported days of 60 min of PA per week and BMI z-score, PACER, and the three environmental factors (community access to PA, school PE, and school PA). A multiple regression model was used to identify jointly significant predictors of self-reported days of 60 min of PA per week. Chi-square tests were performed to assess the relationships between sex, race, weight category, school- and county-level determinants, and whether participants took part in at least one, five, or seven days of 60 min of PA per week. All the analyses were performed using SAS 9.4 (SAS Institute Inc., Cary, NC, USA). A significance level of 0.05 was adopted for all statistical tests.

## 3. Results

Data from a total of 3799 participants from 40 middle schools were included in the final analysis. Across all schools, nearly half of the participants were male (51.1%), half were white (49.3%), and half had a healthy weight (50.7%). Participant characteristics are shown in Table 2.

Figure 1 presents the differences in the mean number of self-reported days of 60 min of PA based on the sex, race, and weight categories. Boys reported significantly more days per week of 60 min of PA than girls (*p* < 0.0001). White participants reported significantly more days of 60 min of PA than black students and students of other races (*p* < 0.0001). Healthy weight participants reported the most days per week of 60 min of PA, while obese students reported the least (*p* < 0.0001).

Spearman correlations between the three environmental factors (community access to PA, school PE, and school PA) and self-reported PA were very weak (|r| < 0.1), although some of the correlations were statistically significant. Self-reported days of 60 min of PA per week and PACER had a stronger and significant positive correlation (r = 0.27, *p* < 0.0001).

The percentage of participants reporting 60 min of self-reported PA on one, five, or seven day(s) per week were compared by school PE, school PA opportunities, access to PA in the community, and sex, race, and weight categories (Table 3).

The percent of participants who reported 60 min of PA in at least five or seven days per week was significantly different across categories of school PE scores and school PA opportunities. The percentage of students who reported 60 min of PA for five (*p* < 0.0001) or seven (*p* = 0.0307) days per week tended to be higher with increased days per week of school PE. The percent of participants who reported 60 min of PA in at least one or five days per week was significantly different across categories of community access to PA.

For the demographic factors, the percent of participants who reported 60 min of PA on at least one, five or seven days per week was significantly higher for boys (vs. girls) and for white students (vs. black and other). The percent of participants who reported 60 min of PA in at least five or seven days per week was significantly higher for healthy weight students comparing to students in other weight categories.

Results from the multiple regression model are shown in Table 4. Race (*p* < 0.0001), school PE (*p* = 0.0011), sex (*p* < 0.0001), and weight category (*p* < 0.0001) were found to be significant predictors of self-reported PA. Male, white, and healthy weight students were found to report significantly more days of 60 min of PA than their counterparts.

## 4. Discussion

Findings from this study reveal that most adolescents that participated in MATCH for the 2018–2019 school year did not get the recommended amount of PA, with a self-reported PA mean of 3.7 days. Furthermore, female, black, underweight, overweight, and obese students were found to report significantly less days of 60 min of PA than their peers. An important finding of this study is that a higher percentage of students reporting more days of PA is correlated with schools providing more frequent PE. These findings may be used to consider possible opportunities at school and community levels to improve PA in adolescents.

The American Academy of Pediatrics recommends that children over age 6 and adolescents receive at least 60 min of PA daily [21]. However, studies consistently show that adolescents do not get this much PA, as we found in this study. Additionally, it has been shown that PA levels decline from childhood through adolescence and adulthood [22]. This can be due to increased demands at school in the high school years, and with employment as teenagers begin to work outside of school. There are also fewer opportunities for organized PE in high school compared with elementary and middle school. As is also consistent with our results, female and minority children obtaining less physical activity is a pervasive theme across the literature. The causes for lower female adolescent participation are multifactorial, but may be partly explained by perceived barriers that affect girls more than boys. Such barriers described in Rosseli et al include social influence and a lack of time, energy, willpower, or skill, and were found to be independent predictors of the estimated total volume of physical activity measured in metabolic equivalents of hours/week [23]. Lack of time, lack of motivation, and self-esteem have been reported as barriers to PA in adolescent girls in some US studies as well [24,25]. For instance, girls may choose to avoid physical activity due to teasing, bullying, and other harmful social interactions [26]. Globally, differences by gender can in part be linked to perception and traditional societal roles for women [27]. 

Additionally, minority racial populations, especially minority female populations, reporting less PA is similarly due to a combination of numerous factors, including socioeconomic status, increased screen time (specifically TV watching), and cultural beliefs across minority groups. [28]. Notably, it has been shown that rural schools located in poorer, more racially heterogeneous regions of North Carolina had less environmental support towards PA than schools located in the least racially heterogeneous areas of the state [4]. A national study by Gordon-Larsen et al found that the odds of having at least one PA facility in a census-block decreased as the minority population increased [29]. Public facilities, youth organizations, schools, and YMCAs were significantly more likely to be in higher-socioeconomic status, low minority areas [29]. This study also found that the relative odds of being overweight declined with increasing numbers of PA facilities per block group [29]. Individuals who lived in census-block groups with more PA facilities were less likely to be overweight and more likely to be highly active than those who lived in block groups with no PA facilities [29]. These findings suggest that minority, overweight, and obese students may engage in less PA due to the lack of recreational resources in their communities. Being underweight in adolescence can also cause several health problems such as scoliosis, pubertal delay, osteoporosis, a weakened immune system, and poor body image perception [30]. In addition to insufficient PA being associated with being overweight in adolescence, it has also been associated with being underweight in adolescence [31,32]. Taken together, physical inactivity in these groups is a topic that warrants further investigation, but this underscores the importance of ensuring an adequate PE provision in a reliable environment for adolescents, such as in schools.

These studies have also demonstrated that although these adolescent populations typically report less PA, one potential way to engage these populations in PA is through school-based PE and PA opportunities. Our study revealed that increased provision of school PE was positively correlated with 60 min of self-reported PA on 5 or 7 days per week and was an independent predictor of the number of self-reported days of 60 min of PA. Policy-based interventions that aim to increase PE and PA in schools can be an important and effective step towards increasing adolescent PA levels, since many adolescents spend a significant part of their day in school. These opportunities can target PE classes, recess, in-class activities, and after-school activities. Although the importance of PA in this age group has been recommended and understood, in the US there are no federal standards or mandates regarding the number of days and/or minutes of PE each week [33]. This general lack of government standard requirement for PE may be due in part from an unintended outcome of federal legislation to address academic achievement, such as the “No Child Left Behind Elementary and Secondary Act” from 2001. This act focused on ensuring academic progress for all students, regardless of background, by allowing states to set curriculum standards [34]. The act is based on four principals of educational reform, with the intent of ensuring that all students reach the same level of academic competency by holding schools accountable for results [35]. Academic content and achievement standards in reading, mathematics, and science were defined by each state, and each state determined what each student should know. State assessments were utilized as the way schools proved that they successfully taught their students. Consequently, schools and districts were held accountable for student outcomes, resulting in more resource diversion to core skills, such as reading and math, to achieve better outcomes on testing. This effort to improve achievement has been noted to have inadvertently led schools to divert time from PE [36]. However, since then, there have been steps towards emphasizing the importance of PE and PA in schools. In 2015, the U.S Congress passed the “Every Student Success Act” (ESSA). The ESSA replaced the “No Child Left Behind Elementary and Secondary Act” and notes that health and PE are key components of a “well-rounded education” [37]. This helps ensure the importance of physical health and fitness is prioritized in schools; however, it leaves room for interpretation on how to effectively enact school-based policies focused on increasing PE. Our study adds support to the current literature for policies in schools to implement strategies to improve time for and access to PE, with an additional focus on female and minority populations.

Lastly, the impact of severe acute respiratory syndrome coronavirus 2 (SARS-CoV-2), the virus responsible for the COVID-19 pandemic, on adolescent PA levels cannot be ignored. Prior to the pandemic most adolescents in the United States were not meeting the recommendations for PA. Mandated school closures, closure of recreational facilities, and stay at home orders made it even more difficult for adolescents to access PA. A systematic review by Mayra et al examined the current literature findings on the impact of the COVID-19 pandemic on PA levels in adolescents and children in the United States [38]. A study by Beck et al. found that daily PA was reduced from 1.8 to 1.0 h per day in children 4–12 years old who were overweight or suffering from obesity [39]. Two of the studies, Tulchin-Francis et al and Dunton et al, found that older children (9–18 years) had a greater decrease in PA compared to younger children (3–8 years) [40,41]. Finally, nearly 80% of a sample of 2440 elementary and middle school PE teachers, school administrators, and school nurses reported significantly less or somewhat less PA during COVID-19 school closures [42]. As a result there, has been a decrease in youth cardiovascular fitness, a marker of physical fitness measured by maximal oxygen intake [43]. Now that mandates for school closures have been lifted, public health interventions are urgently needed to promote participation in PA and PE in schools among adolescents to mitigate the deleterious impact of COVID-19 on PA and physical fitness in the youth.

We should also consider the impact of the pandemic on academic disruption for students. Analysis of student learning through the pandemic by McKinsey revealed that, on average, K-12 students were four months behind in mathematics and three months behind in reading at the start of the 2021–2022 academic year [44]. Their analysis also determines that, in a typical classroom of 30 students, 3 additional students will be 2 or more grades behind for the 2021–2022 academic year [44]. In an effort to support student academic recovery the federal government has committed more than $200 billion to K-12 education through the Elementary and Secondary School Emergency Relief (ESSER) and Governor’s Emergency Education Relief (GEER) funds, with 90% of the funds going to school districts [45]. Via data collected from Burbio’s district report, 1420 districts in 44 states have committed 67% of these funds [44]. Of that amount, 28% has been committed to academic recovery. Although districts have a broad discretion on how to allocate these funds, their priorities are focused on mathematics and English language arts [46,47]. However, based on the findings of the impact of the COVID-19 pandemic on adolescent PA levels, we must support students’ academic progress without sacrificing time towards exercise in the school setting. This is especially true, since there is evidence to suggest that improved academic performance may be related to increasing physical fitness and PA, and time during the school day dedicated to PA [33]. 

### Limitations

Limitations of this study include that the data are from a subset of adolescents from one state in the southeast US and may not be generalizable to other ages or regions. Analyses are cross-sectional, so conclusions cannot be made about cause and effect, and PA levels were self-reported and not objectively measured. Although the survey questions assessing PA are validated and questions had specific instructions, the questions may have been subjected to interpretation by the adolescents filling the questionnaire.

## 5. Conclusions

Obesity and inadequate PA are compelling public health concerns and, thus, increasing youth PA through school-based efforts is a way to help combat these problems. Based on the results of our study, there is opportunity to strengthen efforts to increase PA in adolescents through a focus on increasing PA and PE opportunities in schools, especially making sure that these opportunities are inclusive and mindful of minority and female populations. These efforts may be especially needed in rural locations with increased barriers towards PA. Policy changes that target increasing PA and PE in middle schools may provide a mechanism for schools to contribute to the solutions of improving health for youth, reducing disparities in PA levels, and addressing obesity. The impact of COVID-19 on PA levels and the physical fitness of adolescents increases the importance of these public health interventions.

## Figures and Tables

**Figure 1 ijerph-19-11417-f001:**
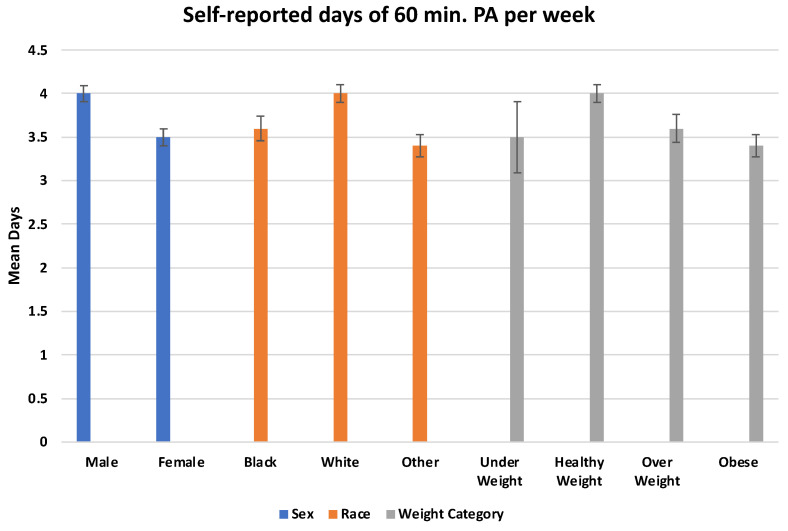
Mean self-reported days of 60 min physical activity (PA) per week by group, with comparison by *t*-test for sex (*p* < 0.0001), and ANOVA for race (*p* < 0.0001), and weight (*p* < 0.0001) categories.

**Table 1 ijerph-19-11417-t001:** Details of how county- and school-level determinants were measured.

Determinant	Description	Level	Scale	Description	Source
County-level Determinant					
Exercise Opportunities	Adequate access to locations for physical activity	County	1–5	“Access to Exercise Opportunities measures the percentage of individuals in a county who live reasonably close to a location for physical activity. Locations for physical activity are defined as parks or recreational facilities. Individuals are considered to have access to exercise opportunities if they: reside in a census block that is within a half mile of a park, reside in an urban census block that is within one mile of a recreational facility, or reside in a rural census block that is within 3 miles of a recreational facility” [17]	County Health Rankings and Roadmaps
School-level Determinants					
Physical Education (PE)	How health/PE is provided for 7th grade students	School	1–5	1 = Not provided in 7th grade 2 = Provided for a subset of students in one semester or block schedule 3 = Provided for a subset of students throughout the year 4 = On a semester or block schedule for all students 5 = Provided for both semesters or throughout the school year for all students	School Administrator Survey
School Physical Activity (PA) Opportunities	PA opportunities available at school outside of regular PE time	School	1, 3, 5	1 = No 3 = Yes, but how this occurs varies by teacher or class 5 = Yes, there is time set aside with organized, appropriate activities and/or places for activity (such as playground, gym, all purpose room, game field, or a place to walk)	School Administrator Survey

**Table 2 ijerph-19-11417-t002:** Characteristics of 2018–2019 MATCH participants and baseline BMI z-score, fitness testing (PACER), and self-reported PA.

Characteristics of 2018–2019 MATCH Participants		
Sample Size, N	Participants	3799
	Schools	40
		N (%)
Sex	Male	1942(51.1%)
	Female	1857 (48.9%)
Ethnicity	Black	942 (24.8%)
	White	1871 (49.3%)
	Other	986 (26.0%)
Weight Status (Based on BMI Percentile)	Underweight (<5%)	108 (2.8%)
	Healthy weight (5–84.9%)	1925 (50.7%)
	Overweight (85–94.9%)	709 (18.7%)
	Obese (≥95%)	1057 (27.8%)
Body Mass Index, Fitness Testing, and Self-Reported PA		
	Mean (SD)	
BMI z-score	0.8 (1.2)	
PACER test result, # of laps	28.7 (20.6)	
Baseline # of days with 60 min PA	3.7 (2.2)	

Abbreviations are as follows: BMI, body mass index; MATCH, motivating adolescents with technology to choose health; PACER, progressive aerobic cardiovascular endurance run; PA, physical activity; N, number of participants; SD, standard deviation.

**Table 3 ijerph-19-11417-t003:** Count (percent) of participants reporting 60 min of self-reported PA on at least one, five or seven day(s) per week) by schools’ PE and PA opportunities, sex, ethnicity, and weight categories.

Variable	N	≥1 Day *	≥5 Days *	7 Days *
School PE				
2	607	542 (89.3%)	209 (34.4%)	74 (12.2%)
3	439	386 (87.9%)	174 (39.6%)	63 (14.4%)
4	1147	1033 (90.1%)	435 (37.9%)	175 (15.3%)
5	1604	1470 (91.7%)	726 (45.2%)	275 (17.1%)
*p*-value		0.1618	<0.0001	0.0307
School PA				
1	1631	1454 (89.2%)	637 (39.1%)	231 (14.2%)
3	1336	1224 (91.6%)	596 (44.6%)	233 (17.4%)
5	830	753 (90.7%)	311 (37.5%)	123 (14.8%)
*p*-value		0.0707	0.001	0.0414
Access to PA				
1	309	295 (95.5%)	164 (53.1%)	56 (18.1%)
2	703	634 (90.2%)	263 (37.4%)	112 (15.9%)
3	711	642 (90.3%)	277 (39.0%)	101 (14.2%)
4	1428	1261 (88.3%)	519 (36.3%)	206 (14.4%)
5	646	599 (92.72)	321 (49.7%)	112 (17.3%)
*p*-value		0.0004	<0.0001	0.2342
Sex				
Female	1857	1643 (88.5%)	667 (36.0%)	232 (12.5%)
Male	1940	1788 (92.2%)	877 (45.2%)	355 (18.3%)
*p*-value		0.0001	<0.0001	<0.0001
Race				
Black	942	823 (87.4%)	357 (37.9%)	149 (15.8%)
White	1871	1720 (92.0%)	874 (46.7%)	327 (17.5%)
Other	984	888 (90.2%)	313 (31.8%)	111 (11.3%)
*p*-value		0.0006	<0.0001	<0.0001
Weight Category				
Under Weight	108	94 (87.0%)	33 (30.6%)	15 (13.9%)
Healthy Weight	1924	1756 (91.3%)	884 (46.0%)	344 (17.9%)
Overweight	709	630 (88.9%)	273 (38.5%)	101 (14.3%)
Obese	1056	951 (90.1%)	354 (33.5%)	127 (12.03%)
*p*-value		0.1618	<0.0001	0.0003

Abbreviations are as follows: PE, physical education; PA, physical activity. * Percentages in each row add up to more than 100% because the categories are not mutually exclusive.

**Table 4 ijerph-19-11417-t004:** Multiple regression model between school PE, sex, ethnicity, weight category, and self-reported days per week (0–7) of 60 min of PA.

Variable	Estimate (B)	Standard Error	*p*-Value
School PE	0.108	0.033	0.0011
Sex			<0.0001
Female	−0.526	0.070	<0.0001
Male	-	-	-
Race			<0.0001
Black	−0.306	0.088	0.0005
Other	−0.542	0.086	<0.0001
White	-	-	-
Weight Category			<0.0001
Under Weight	−0.638	0.213	0.0028
Overweight	−0.281	0.095	0.0030
Obese	−0.477	0.083	<0.0001
Healthy Weight	-	-	-

Abbreviations are as follows: PE, physical education; PA, physical activity.

## Data Availability

The data files will be made available upon request.

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
