# Peer review of "Determining Trends and Factors Associated with Self-Reported Physical Activity among Adolescents in Rural North Carolina"

_ijerph, 2022, doi:10.3390/ijerph191811417_

Round 1

Reviewer 1 Report

Overall – This study is an interesting piece of work that corroborates previous evidence regarding the barriers and facilitators to meet recommendations about moderate-to-vigorous physical activity among adolescents. It will add worthwhile evidence, which might support the implementation of school policies aimed to improve PA levels among all students. 

Introduction – The introduction section is well written and provides a strong rationale for this study. 

Methods – The methods are clearly explained. 

Results – The results section is comprehensive and describes all the findings that the authors decided to investigate.

One minor point:

Page 6 – In the last paragraph of the Results sessions, the authors mentioned Table 4, but there is no such table. I believe they were referring to Table 3. 

Discussion – The discussion links well with the introduction and the research question. The focus on PE in the third paragraph is essential. As the authors mentioned, there is a need to support, with evidence, the implementation of new strategies to improve school access to PE classes.  

My only concern regards the lack of explanation about why “black, underweight, overweight, and obese students” engage in less PA (first paragraph in the Discussion).

The authors, in the second paragraph of the discussion, explained very well the potential reasons why female students engage in fewer PA levels than males. However, although this study also shows that black, underweight, overweight, and obese students engage in fewer days of PA than their peers, no potential explanations are given regarding the reasons why these populations engage in less PA. 

Despite the authors mentioning minority students, it is not clear if they were referring to “black, underweight, overweight, and obese students” or other minorities. Perhaps, it would be beneficial to further investigate the reasons why black, underweight, overweight, and obese students engage in less PA than their peers and add another paragraph to the discussion section.

Limitation – The limitations were addressed and acknowledged. 

References – This study is supported by adequate references. 

Reviewer 2 Report

This manuscript, titled "Determining Trends and Factors Associated with Self-Reported Physical Activity among Adolescents in Rural North Carolina" examined individual, school, and county level factors associated with participation in physical activity in a large sample of seventh grade students in a rural area of North Carolina. I commend the authors on recruiting such a large representative sample. However, there are some fatal flaws in the presentation of the methods and results as currently presented. Although the analysis is simple, inappropriate analysis techniques were chosen for the data. Additionally, the way that the results were presented were confusing and very hard to follow at times. I have more specific comments below for the authors to consider to increase the publishability of their study.

Introduction

-        Please refer to a theoretical model that underlies your research. The ecological model for active living is a good example that recognizes that there are multiple levels of influence on physical activity including individual, environmental, social, and policy factors.

-        The authors discuss a pedometer-based intervention to increase physical activity in school children. I don’t see the relevance of this to the study described in the manuscript. I would suggest removing.

-        Please provide specific study hypothesis or clarify that this is an exploratory study.

Methods

-        The first paragraph of the methods section seems redundant to me given each of the dependent and independent variables are described in detail later in the methods section. I would suggest removing.

-        Please clarify who rated the county level predictor “exercise opportunity”. It is also not clear what the points on the 1-5 scale correspond to.

-        Please rename the school-level determinant “physical activity (PA)” in Table 1. It’s very confusing to have an independent and dependent variable that share the same name.

-        Please provide more detail about the PACER test. How long is a completed lap? How much time are students given?

-        It’s not appropriate to treat the school and county level independent variables as continuous variables to calculate a correlation coefficient. They are clearly not continuous variables. Please use ANOVAs to compare hours of physical activity between each of the categories of school and county independent variables rather than correlations.

Results

-        The in-text descriptive statistics are redundant because they are just repeating information that is available in Table 2. This information could be removed and readers just need to be referred to the table.

-        Below Table 2 it says “Testing (PACER), and Self-Reported PA.” There is no context to this. Why is it there?

-        Please add 95% confidence interval error bars to Figure 1 to aid interpretation.

-        Please remove correlations as mentioned in my comment above.

-        I find Table 3 very confusing to read. What do the p-values signify? The percentages don’t add up to 100% in rows or columns. Why do cells with lower counts have greater percentages? Also, the categories are odd to me. What is a participant had zero, 2, 3, 4 or 6 days of 60 minutes of physical activity? Were they not included in the table?  
